# On Margins and Generalisation for Voting Classifiers

**Felix Biggs**
Department of Computer Science
University College London and Inria
London
contact@felixbiggs.com

**Valentina Zantedeschi**
ServiceNow Research,
University College London and Inria
London
vzantedeschi@gmail.com

**Benjamin Guedj**
Department of Computer Science
University College London and Inria
London
b.guedj@ucl.ac.uk

## Abstract

We study the generalisation properties of majority voting on finite ensembles of classifiers, proving margin-based generalisation bounds via the PAC-Bayes theory. These provide state-of-the-art guarantees on a number of classification tasks. Our central results leverage the Dirichlet posteriors studied recently by Zantedeschi et al. (2021) for training voting classifiers; in contrast to that work our bounds apply to non-randomised votes via the use of margins. Our contributions add perspective to the debate on the "margins theory" proposed by Schapire et al. (1998) for the generalisation of ensemble classifiers.

## 1 Introduction

Weighted ensemble methods are among the most widely-used and effective algorithms known in machine learning. Variants of boosting (Freund and Schapire, 1997; Chen and Guestrin, 2016) are state-of-the-art in a wide variety of tasks (Shwartz-Ziv and Armon, 2022; Nielsen, 2016) and methods such as random forest (Breiman, 2001) are among the most commonly-used in machine learning competitions (see, *e.g.*, Bell and Koren, 2007; Uriot et al., 2021), valued both for their excellent results and interpretability. Even when these algorithms do not directly produce the best learners for a task, the best performance in competitions is often obtained by an ensemble of "strong learners"—the output of a collection of different algorithms trained on the data—contrasted to the weak learners usually considered in the ensemble learning literature.

Among the oldest ideas to explain the performance of ensemble classifiers, and machine learning methods in general, is the concept of margins. First introduced to analyse the Perceptron algorithm (Novikoff, 1962), margins relate closely to the idea of confidence in predictions in ensemble learning, with a large margin implying that a considerable weighted fraction of voters chose the same answer. This was first leveraged to obtain early margin-based generalisation bounds for ensembles by Schapire et al. (1998), in an attempt to understand the excellent generalisation of boosting, a surprising result given classical Vapnik-Chervonenkis theory. This "margins theory" was explored further in a number of works (Wang et al., 2008; Gao and Zhou, 2013; Grønlund et al., 2020) and is among the leading explanations for the success of such methods and boosting in particular.

The same thread of margin bounds for ensemble methods has also been taken up in parallel in PAC-Bayes theory by Langford and Seeger (2001); Biggs and Guedj (2022b). PAC-Bayes provides a natural framework both for deriving margin bounds, and for considering ensemble methods in general, particularly majority votes where the largest-weighted ensemble prediction is taken. Within

the framework, the weightings are typically considered as the parameter of a categorical distribution over individual voters. PAC-Bayes theorems (see the comprehensive surveys of Guedj, 2019; Alquier, 2021) then directly provide generalisation bounds for the performance of this "randomised" proxy for the majority vote, *a.k.a.* Gibbs classifier. These can then be de-randomised by such margin-based techniques, or through a variety of oracle bounds (Langford and Shawe-Taylor, 2003; Shawe-Taylor and Hardoon, 2009; Lacasse et al., 2010; Masegosa et al., 2020), motivating new learning algorithms (Lacasse et al., 2006; Roy et al., 2011; Germain et al., 2015; Laviolette et al., 2017; Lorenzen et al., 2019; Viallard et al., 2021; Wu et al., 2021).

Uniquely among PAC-Bayesian approaches, Zantedeschi et al. (2021) instead consider Dirichlet distributions over the voters. Any sample from this distribution already implies a vector of voting weights, and it is on the performance and optimisation of these "stochastic majority votes" they primarily focus. As an aside, they provide an oracle result which allows their bounds to be de-randomised, but this introduces an irreducible factor such that the bound on the true fixed vote can never be less than double that of the stochastic version. It also neglects to leverage the generally high confidence of predictions obtained by their algorithm.

**Our contribution.** By combining tools from margin bounds and the use of Dirichlet majority votes, we provide a new margin bound for non-randomised majority votes. This is in contrast to Zantedeschi et al. (2021) which primarily considers stochastic majority votes. Our bound empirically compares very favourably to existing margin bounds and in contrast to them are applicable to multi-class classification. Remarkably, our empirical results are also sharper than existing PAC-Bayesian ones, even when the algorithm optimising those bounds is used.

Our primary tool is a new result relating the margin loss of these stochastic votes to the misclassification loss of the non-randomised ones in a surprisingly sharp way. This tool can additionally be utilised alongside a further idea from Zantedeschi et al. (2021) to obtain an alternative form of the bound which is more amenable to optimisation. Through this work we provide further support to the margins theory for ensembles, showing that near-sharp bounds based on margins alone can be obtained on a variety of real-world tasks.

**Outline.** The rest of this section introduces the problem setup, notation and summarises main results. Section 2 provides background on PAC-Bayes, Dirichlet majority votes and margin bounds, relating them to our new results. Section 3 states and summarises our new theoretical results, giving the most relevant proofs (all remaining proofs are deferred to appendices). Section 4 empirically evaluates these new results before we conclude with an overall discussion in Section 5.

## 1.1 Notation and setting

Majority voting algorithms combine the predictions of a finite set of "base" classifiers, $\mathcal{H}$, from $\mathcal{X}$ to $\mathcal{Y} = [c] := \{1, \ldots, c\}$. The classifiers $h_i \in \mathcal{H}$ take the form $h_i : \mathcal{X} \to \mathcal{Y}$ for $i \in [d]$ so that $|\mathcal{H}| = d$. Majority votes consider as set of weightings $\boldsymbol{\theta}$ in $\Delta^d$, the simplex, and return the highest-weighted overall prediction. Using the indicator function $\boldsymbol{I}_{\mathcal{A}}$ of a set $\mathcal{A}$, this is expressed as

$$f_{\boldsymbol{\theta}}(x) = \underset{k \in \mathcal{Y}}{\operatorname{argmax}} \sum_{i \in [d]} \theta_i \boldsymbol{I}_{h_i(x)=k}.$$

We are primarily interested in learning a weighting $\boldsymbol{\theta}$ with small misclassification risk (and guarantees of this) based on a sample $S \sim \mathcal{D}^m$, where $\mathcal{D} \in \mathcal{M}_1^+(\mathcal{X} \times \mathcal{Y})$ is the data-generating distribution and $m \in \mathbb{N}_+$ the sample size. We let $\mathcal{M}_1^+(\mathcal{A})$ denote the set of probability measures on a set $\mathcal{A}$. For $h \in \mathcal{H}$ the misclassification loss is $\ell_0(h, x, y) := \boldsymbol{I}_{h(x) \neq y}$, the misclassification out-of-sample risk is $L_0(h) := \mathbb{E}_{(x,y) \sim \mathcal{D}} \ell_0(h, x, y)$ and a hat denotes the in-sample estimate of this quantity, $\hat{L}_0(h) := \mathbb{E}_{(x,y) \sim \text{Uniform}(S)} \ell_0(h, x, y)$. In a slight abuse of notation we will also often write the risk of the majority vote $L_0(\boldsymbol{\theta}) = L_0(f_{\boldsymbol{\theta}})$ and similarly for its empirical counterpart.

The margin of majority vote $f_{\boldsymbol{\theta}}$ on example $(x, y)$ is derived from the minimal gap between the total weight assigned to the true class $y$ and to any other predicted class:

$$M(\boldsymbol{\theta}, x, y) := \frac{1}{2} \sum_{i : h_i(x)=y} \theta_i - \frac{1}{2} \max_{k \neq y} \sum_{i : h_i(x)=k} \theta_i.$$

The corresponding margin loss is $\ell_\gamma(\boldsymbol{\theta}, x, y) := \boldsymbol{I}_{M(\boldsymbol{\theta}, x, y) \leq \gamma}$ for margin $\gamma \geq 0$, with the corresponding in-sample and out-of-sample risks notated as $\hat{L}_\gamma(\boldsymbol{\theta})$ and $L_\gamma(\boldsymbol{\theta})$ respectively.

## 1.2 Overview of results

Our main result is a margin bound of the following form: with high probability $\geq 1 - \delta$ over the sample, simultaneously for any $\boldsymbol{\theta} \in \Delta^d$ and $K > 0$,

$$L(\boldsymbol{\theta}) \leq O\left(\hat{L}_\gamma(\boldsymbol{\theta}) \;+\; e^{-K\gamma^2} \;+\; \frac{\mathbb{D}_{\mathrm{Dir}}(K\boldsymbol{\theta}, \mathbf{1}) + \log \frac{m}{\delta}}{m}\right) \tag{1}$$

where $\mathbb{D}_{\mathrm{Dir}}(\boldsymbol{\alpha}, \boldsymbol{\beta})$ is the KL divergence between Dirichlet random vectors with parameters $\boldsymbol{\alpha}$ and $\boldsymbol{\beta}$, with $\mathbf{1}$ a vector of ones implying a uniform Dirichlet prior distribution on the simplex. The term $e^{-K\gamma^2}$ is a de-randomisation penalty. The parameter $K$ is chosen freely in an arbitrary data-dependent way to balance the requirements of the different terms: it must be large enough to decrease this exponential term, while too-large a parameter increases the KL divergence from the uniform prior. This result is surprisingly strong; in particular there is no dependence on the dimensionality (*i.e.*, number of voters $d$) in the exponential term, an advantage discussed further in Section 3.2.

In Equation (1), $\hat{L}_\gamma(\boldsymbol{\theta})$ is the 0-1 valued $\gamma$-margin loss which enables comparison with existing margin bounds for trained weighted ensembles. We further consider a second scenario, where the generalization bound is also used to train the model itself. We note that the $\gamma$-margin loss $\hat{L}_\gamma(\boldsymbol{\theta})$ appearing in Equation (1) has null gradients, so the bound cannot be directly optimised by gradient descent. To rectify this we also prove a variation of the bound, replacing the above loss by its expectation under a Dirichlet stochastic vote, $\mathbb{E}_{\boldsymbol{\xi} \sim \mathrm{Dir}(K\boldsymbol{\theta})} \hat{L}_\gamma(\boldsymbol{\xi})$, which is bounded in differentiable closed form to give an alternative, optimisation-friendly bound.

In our evaluations we focus on these two complementary scenarios, obtaining state-of-the-art empirical results. Across different scenarios and tasks our results outperform both existing margin bounds (including a sharpened version of the result from Biggs and Guedj (2022b) which may be of independent interest), and PAC-Bayes bounds, even when it is not used as the objective. Further, in contrast to existing margin bounds our results also hold for multi-class majority votes.

# 2 Background

## 2.1 PAC-Bayes bounds

PAC-Bayes bounds are among the tightest known generalisation bounds, as for example the only framework in which non-vacuous generalisation bounds for neural networks have been obtained (see *e.g.* Dziugaite and Roy, 2017, 2018; Zhou et al., 2019; Letarte et al., 2019; Dziugaite et al., 2021; Perez-Ortiz et al., 2021; Biggs and Guedj, 2021, 2022a). However, unlike many other such bounds they usually apply to randomised Gibbs(-like) prediction functions rather than deterministic ones. These are typically re-drawn for every new test evaluation. Thus a high-probability bound is obtained on the expectation of the risk w.r.t. the PAC-Bayes posterior $Q$, with the complexity of $Q \in \mathcal{M}_1^+(\mathcal{H})$ appearing in the bound in terms of a Kullback-Leibler (KL) divergence from a pre-chosen PAC-Bayes prior $P \in \mathcal{M}_1^+(\mathcal{H})$ (which is not required to be a true prior in the Bayesian sense – see the discussion in Guedj, 2019). A particularly sharp (as discussed in Foong et al., 2021) and widely-used result is given in Theorem 1, valid for any bounded loss function $\ell$ with values in $[0, 1]$.

**Theorem 1** (Seeger et al. (2001); Maurer (2004)). *For any $\mathcal{D} \in \mathcal{M}_1^+(\mathcal{X} \times \mathcal{Y})$, $m \in \mathbb{N}_+$, prior $P \in \mathcal{M}_1^+(\mathcal{H})$ and $\delta \in (0, 1)$, with probability $\geq 1 - \delta$ over $S \sim D^m$, simultaneously for all $Q \in \mathcal{M}_1^+(\mathcal{H})$*

$$\mathbb{E}_{h \sim Q} L(h) \leq \mathrm{kl}^{-1}\left(\mathbb{E}_{h \sim Q} \hat{L}(h), \; \frac{1}{m}\left(\mathrm{KL}(Q, P) + \log \frac{2\sqrt{m}}{\delta}\right)\right)$$

*where the generalised inverse* $\mathrm{kl}^{-1}(u, c) := \sup\{v \in [0, 1] : \mathrm{kl}(u, v) \leq c\}$ *and* $\mathrm{kl}(u, v) := u \log \frac{u}{v} + (1 - u) \log \frac{1-u}{1-v}$ *is a KL divergence between Bernoulli random variables.*

The above bound uses the inverse small-kl function which will be seen in our later results and a number of pre-existing ones. To lend intuition we note that $\mathrm{kl}^{-1}(u, c) \in O(u + c)$, giving

Equation (1) from Theorem 2 when using a uniform prior. The following upper bounds are also useful: $\mathrm{kl}^{-1}(u, c) \le u + \sqrt{c/2}$ giving "slow-rates" and $\mathrm{kl}^{-1}(u, c) \le u + \sqrt{2cu} + 2c$. From this we can see that when the loss $\hat{L} \to 0$ then the overall rate improves to $O(1/m)$, so the small-kl formulation interpolates between the traditional fast and slow rate regimes of learning theory.

## 2.2 Margin bounds

In the learning theory literature there exists a rich tradition of using the concept of a margin, which quantifies the confidence of predictions, to explain generalisation. This is particularly evident in the case of voting algorithms such as boosting, where traditional Vapnik-Chervonenkis based techniques predict classical overfitting which is not ultimately observed. The "margins theory" was developed by Schapire et al. (1998) to explain this discrepancy. By considering the weightings $\boldsymbol{\theta}$ as the parameter of a categorical distribution, they proved a bound of the form (holding with probability greater than $1 - \delta$ over the sample, as for all bounds in this section) $L_0(\boldsymbol{\theta}) \le L_\gamma(\boldsymbol{\theta}) + \tilde{O}\left(\frac{1}{\gamma\sqrt{m}}\right)$. Although there was initially some debate about the validity of the theory (Breiman, 1999), eventually Gao and Zhou (2013, Theorem 4) provided the following improved bound which further supported that a large-margin voting classifier could generalise: simultaneously for any $\gamma > \sqrt{2/d}$ and $\boldsymbol{\theta} \in \Delta^d$,

$$L_0(\boldsymbol{\theta}) \le \mathrm{kl}^{-1}\left(\hat{L}_\gamma(\boldsymbol{\theta}), \; \frac{1}{m}\left(\frac{2\log(2d)}{\gamma^2}\log\frac{2m^2}{\log d} + \log\frac{dm}{\delta}\right)\right) + \frac{\log d}{m}. \tag{2}$$

More recently, a similar bound (proved through a PAC-Bayesian method based on Seeger et al. (2001)) was proved in Biggs and Guedj (2022b, Theorem 8). Here we give a bound provided as an intermediate step in their proof that is strictly (and empirically considerably) sharper than their final result: for any fixed margin $\gamma > 0$, simultaneously for any $\boldsymbol{\theta} \in \Delta^d$

$$L_0(\boldsymbol{\theta}) \le \mathrm{kl}^{-1}\left(\hat{L}_\gamma(\boldsymbol{\theta}) + \frac{1}{m}, \; \frac{1}{m}\left(\lceil 2\gamma^{-2}\log m\rceil \log d + \log\frac{2\sqrt{m}}{\delta}\right)\right) + \frac{1}{m}. \tag{3}$$

Since $\gamma \in (0, \frac{1}{2})$ for non-vacuous results, a union bound argument can be used to extend the above to fixed-precision $\gamma$, and this result has the advantage of being valid for small $\gamma$ as are often observed empirically.

**Our contributions.** Firstly we mention the smaller contribution of the improved form of the bound from Biggs and Guedj (2022b) given in Equation (3); a proof is given in Appendix B alongside further refinements and evaluation. However we show that in many cases even this improved version and Equation (2) give weak or vacuous results. As a result of this weakness (and thus perhaps null result for the margins theory applied to voting classifiers) we present a new margin bound in Theorem 2 based on Dirichlet distributions as a theoretical intermediate step. This is also valid in the multi-class case, unlike the above results which are only for binary classification. Empirically the bound is observed to give an enormous improvement in tightness than the existing margin bounds and in some cases is near-sharp.

## 2.3 Dirichlet stochastic majority votes

In most results from the PAC-Bayes framework, and in the proof of the existing results given in Section 2.2, the majority vote weightings $\boldsymbol{\theta}$ are considered the parameters of a categorical distribution over voters. Zantedeschi et al. (2021) instead consider PAC-Bayesian bounds (specifically, Theorem 1) applied to a hypothesis class of majority votes of the form $f_{\boldsymbol{\xi}}$, where $\boldsymbol{\xi} \sim \mathrm{Dir}(\boldsymbol{\alpha})$ is drawn from a Dirichlet distribution with parameter $\boldsymbol{\alpha}$. This distribution has mean $\mathbb{E}\boldsymbol{\xi} = \boldsymbol{\alpha}/\sum_{i=1}^d \alpha_i$ with a larger sum $\sum_{i=1}^d \alpha_i$ giving a more concentrated or peaked distribution (see Appendix A for more details).

Since $\boldsymbol{\xi}$ is randomised, the bounds from Zantedeschi et al. (2021) apply to "stochastic majority votes" rather than the more typical deterministic ones we consider here. However, the use of such Dirichlet distributions over voters in the PAC-Bayes bounds rather than the more usual categorical ones is a major step forward as it allows the correlation between voters to be more carefully considered. This is because with a categorical distribution, the expected Gibbs risk is simply an average of the losses of individual predictors, without taking into account how well the combination of their predictions performs. Conversely, the Dirichlet distribution gives a (stochastic) majority vote of predictors, so if

the errors of base voters are de-correlated, the better performance that results from their combination can be accounted for in the bound. We will utilise and de-randomise these stochastic majority votes as a stepping stone to bounds for deterministic predictors $f_{\boldsymbol{\theta}}$ directly.

As is common in the PAC-Bayes literature, Zantedeschi et al. (2021) use their new bound as an optimisation objective to obtain a new algorithm, here using stochastic gradient descent. The bound with Dirichlet posterior obtained directly from Theorem 1 includes the expected misclassification loss with respect to the Dirichlet parameters, $\mathbb{E}_{\boldsymbol{\xi} \sim \operatorname{Dir}(\boldsymbol{\alpha})} \ell(f_{\boldsymbol{\xi}}, x, y)$, which has null gradient for any sampled $\boldsymbol{\xi}$. They therefore additionally upper bound this term by the differentiable closed form

$$\mathbb{E}_{\boldsymbol{\xi} \sim \operatorname{Dir}(\boldsymbol{\alpha})} \ell(f_{\boldsymbol{\xi}}, x, y) \leq I_{\frac{1}{2}} \left( \sum_{i:h_i(x)=y} \alpha_i, \sum_{i:h_i(x) \neq y} \alpha_i \right), \qquad (4)$$

where $I_z(a, b)$ is the regularised incomplete beta function, which has a sigmoidal shape. The inequality is sharp in the binary classification case, and is used in the training objective and final evaluation of their method. As an aside, Zantedeschi et al. (2021) also proved an oracle bound which allows their result to be de-randomised, but this introduces a irreducible factor of two. This bound, which holds with probability at least $1-\delta$ over the sample for any $\boldsymbol{\theta} \in \Delta^d, K > 0$ is given by

$$L_0(\boldsymbol{\theta}) \leq 2 \operatorname{kl}^{-1} \left( \mathbb{E}_{\boldsymbol{\xi} \sim \operatorname{Dir}(K\boldsymbol{\theta})} \hat{L}_0(\boldsymbol{\xi}), \; \frac{\mathbb{D}_{\operatorname{Dir}}(K\boldsymbol{\theta}, \boldsymbol{\beta}) + \log \frac{2\sqrt{m}}{\delta}}{m} \right).$$

**Our contributions.** Firstly, we provide a new margin bound for majority vote algorithms utilising Dirichlet posteriors as a theoretical stepping stone. We show that this bound gives sharper bounds on the misclassification loss than the bound from Zantedeschi et al. (2021), doing better than the irreducible factor, even when applied to the output of their algorithm. We show further that the bound is also tighter when applied to the outputs of other PAC-Bayes algorithms derived from "categorical"-type posteriors. Finally, we give an altered form of the bound involving the expectation of the margin loss $\mathbb{E}_{\boldsymbol{\xi} \sim \operatorname{Dir}(\boldsymbol{\alpha})} \ell_{\gamma}(f_{\boldsymbol{\xi}}, x, y)$ and a result analogous to Equation (4) for this case. Through this we are able to obtain a new PAC-Bayes objective which is compared to existing PAC-Bayes optimisation methods.

## 3 Main results

Our main results use the idea of Dirichlet stochastic majority votes from Zantedeschi et al. (2021) as an intermediate step to prove new margin bounds for deterministic majority votes. In this section, first we give our main result in Theorem 2 and discuss further. In Section 3.1 we give an alternative bound obtained by a very similar method which is more amenable to optimisation, and we provide proofs for these results in Section 3.2.

The central step in these proofs is in constructing a proxy Dirichlet distribution $\boldsymbol{\xi} \sim \operatorname{Dir}(K\boldsymbol{\theta})$ over voters, the loss of which is bounded à la PAC-Bayes, and de-randomised using margins to obtain bounds directly for $f_{\boldsymbol{\theta}}$. The primary complexity term appearing in our bounds is therefore $\mathbb{D}_{\operatorname{Dir}}(K\boldsymbol{\theta}, \boldsymbol{\beta})$, the KL divergence between Dirichlet distributions with parameters $K\boldsymbol{\theta}$ and $\boldsymbol{\beta}$ respectively. As with PAC-Bayes priors, $\boldsymbol{\beta}$ can be chosen in arbitrary sample-independent fashion, but we typically choose it as a vector of ones, giving a uniform distribution on the simplex as prior as in Equation (1). The bounds also involve a de-randomisation penalty of $O(e^{-K\gamma^2})$ where $\gamma$ is the margin appearing in the loss; this term upper bounds the difference between our randomised proxy $\boldsymbol{\xi}$ and its mean $\boldsymbol{\theta}$ and gets smaller with $K$ as the distribution concentrates tightly around its mean. This parameter $K$ can be optimised in any data-dependent way to obtain the tightest final bound.

**Theorem 2.** *For any $\mathcal{D} \in \mathcal{M}_1^+(\mathcal{X} \times \mathcal{Y})$, $m \in \mathbb{N}_+$, margin $\gamma > 0$, $\delta \in (0, 1)$, and prior $\boldsymbol{\beta} \in \mathbb{R}_+^d$, with probability at least $1 - \delta$ over the sample $S \sim \mathcal{D}^m$ simultaneously for every $\boldsymbol{\theta} \in \Delta^d$ and $K > 0$,*

$$L_0(\boldsymbol{\theta}) \leq \operatorname{kl}^{-1} \left( \hat{L}_{\gamma}(\boldsymbol{\theta}) + e^{-(K+1)\gamma^2}, \; \frac{\mathbb{D}_{\operatorname{Dir}}(K\boldsymbol{\theta}, \boldsymbol{\beta}) + \log \frac{2\sqrt{m}}{\delta}}{m} \right) + e^{-(K+1)\gamma^2}.$$

Theorem 2 differs from the existing margin bounds of Equations (2) and (3), and Schapire et al. (1998) in a specific and significant way, with $\boldsymbol{\theta}$ appearing not only in the loss function $\hat{L}_{\gamma}(\boldsymbol{\theta})$, but *also*

in the KL complexity term. Empirically we find our bound to be an improvement but it is possible to generate scenarios where the pre-existing bounds are non-vacuous while ours is not, since the KL divergence is unbounded for certain choices of $\boldsymbol{\theta}$, for example when one of the components is exactly zero. This difference arises because the existing bounds all use the idea of a categorical distribution with parameter $\boldsymbol{\theta}$ in their proofs (which has KL divergence from a uniform prior upper bounded by $\log d$), while we use a Dirichlet. This gains us the surprisingly tight de-randomisation result (Theorem 4) used in all proofs.

## 3.1 PAC-Bayes bound as objective

We note here that it is non-trivial to directly obtain a training objective for optimisation from Theorem 2, due to the non-differentiability of the margin loss $\hat{L}_\gamma(\boldsymbol{\theta})$. Therefore, in order to compare results with a wide variety of methods that optimise PAC-Bayes bounds (including those used by Zantedeschi et al., 2021, as baselines), we obtain a relaxed and differentiable formulation in Theorem 3 for direct optimisation.

**Theorem 3.** *Under the conditions of Theorem 2 the following bound also holds*

$$L_0(\boldsymbol{\theta}) \leq \mathrm{kl}^{-1}\left(\mathbb{E}_{\boldsymbol{\xi}\sim\mathrm{Dir}(K\boldsymbol{\theta})}\hat{L}_\gamma(\boldsymbol{\xi}), \frac{\mathbb{D}_{\mathrm{Dir}}(K\boldsymbol{\theta},\boldsymbol{\beta}) + \log\frac{2\sqrt{m}}{\delta}}{m}\right) + e^{-4(K+1)\gamma^2}.$$

*Using the incomplete Beta function $I_z(a,b)$ we also have the following result, which is sharp in the binary classification case,*

$$\mathbb{E}_{\boldsymbol{\xi}\sim\mathrm{Dir}(\boldsymbol{\alpha})}\ell_\gamma(\boldsymbol{\xi},x,y) \leq I_{\frac{1}{2}+\gamma}\left(\sum_{i:h_i(x)=y}\alpha_i, \sum_{i:h_i(x)\neq y}\alpha_i\right).$$

Theorem 3 has a stronger PAC-Bayesian flavour than Theorem 2, with an expected loss under some distribution appearing (complicating the final optimisation of $K$), while Theorem 2 takes a form much closer to that of a classical margin bound. The second part of the result is analogous to Equation (4) used by Zantedeschi et al. (2021). We combine both parts to calculate the overall bound in closed form and obtain gradients for optimisation.

## 3.2 Proof of main results

The proof of Theorems 2 and 3 essentially follow from applying a simple PAC-Bayesian bound in combination with the key Theorem 4 below. In some sense this is our most important and novel result. Our whole approach is largely motivated by its surprising tightness; in particular there is no dependence on the dimension, which is avoided by careful use of the aggregation property of the Dirichlet distribution. This surprise arises because to obtain a tightly concentrated Dirichlet distribution on $\boldsymbol{\xi} \sim \mathrm{Dir}(\boldsymbol{\alpha})$, the concentration parameter $K = \sum_{i=1}^d \alpha_i$ must grow linearly with the dimension. In fact, even a uniform distribution (which will be less peaked than our final posterior) has $\sum_{i=1}^d \alpha_i = d$, so the de-randomisation step is effectively very cheap in higher dimensions.

**Theorem 4.** *Let $\boldsymbol{\theta} \in \Delta^d$ and $K > 0$. Then for any $\gamma > 0$ and $(x,y)$,*

$$\ell_0(\boldsymbol{\theta},x,y) \leq \mathbb{E}_{\boldsymbol{\xi}\sim\mathrm{Dir}(K\boldsymbol{\theta})}\ell_\gamma(\boldsymbol{\xi},x,y) + e^{-4(K+1)\gamma^2},$$

$$\mathbb{E}_{\boldsymbol{\xi}\sim\mathrm{Dir}(K\boldsymbol{\theta})}\ell_\gamma(\boldsymbol{\xi},x,y) \leq \ell_{2\gamma}(\boldsymbol{\theta},x,y) + e^{-4(K+1)\gamma^2}.$$

For our proofs we first recall the aggregation property of the Dirichlet distribution: if $(\xi_1,\ldots,\xi_d) \sim \mathrm{Dir}((\alpha_1,\ldots,\alpha_d))$, then $(\xi_1,\ldots,\xi_{d-1} + \xi_d) \sim \mathrm{Dir}((\alpha_1,\ldots,\alpha_{d-1} + \alpha_d))$. We further note the following crucial concentration-of-measure result. The aforementioned lack of dimensionality in Theorem 4 is possible because Theorem 5 depends only on $\sum_{i=1}^d \alpha_i$, and this value is unchanged by aggregation, which avoids the dimension dependence that would otherwise be introduced by the requirement $\|\boldsymbol{u}\|_2 = 1$ below.

**Theorem 5** (Marchal and Arbel, 2017). *Let $\boldsymbol{X} \sim \mathrm{Dir}(\boldsymbol{\alpha})$, $t > 0$, and $\boldsymbol{u} \in \mathbb{R}^d$ with $\|\boldsymbol{u}\|_2 = 1$. Then*

$$\mathbb{P}_{\boldsymbol{X}}\{\boldsymbol{u}\cdot(\boldsymbol{X} - \mathbb{E}\boldsymbol{X}) > t\} \leq \exp\left(-2\left(\sum_{i=1}^d \alpha_i + 1\right)t^2\right).$$

*Proof of Theorem 2 and Theorem 3.* The proof of our main results is completed by applying the PAC-Bayes bound Theorem 1 with the $\gamma$-margin loss to a Dirichlet prior and posterior with parameters $\boldsymbol{\beta}$ and $K\boldsymbol{\theta}$ respectively. Substituting the first part of Theorem 4 gives the first part of Theorem 3, and additionally substituting the second part and re-scaling $\gamma \to \gamma/2$ gives Theorem 2.

For the second part of Theorem 3, define $w = \{i : h_i(x) = y\}$ for fixed $(x, y)$ so $W := \sum_{i \in w} \xi_i \sim$ Beta $\left(\sum_{i \in w} \alpha_i, \sum_{i \notin w} \alpha_i\right)$ by the aggregation property of the Dirichlet distribution. Then

$$\mathbb{E}_{\boldsymbol{\xi}} \ell_\gamma(\boldsymbol{\xi}, x, y) \leq \mathbb{E}_{\boldsymbol{\xi}} \left\{ W \geq \frac{1}{2} - \gamma \right\} = 1 - I_{\frac{1}{2} - \gamma} \left( \sum_{j \in w} \alpha_j, \sum_{i \notin w} \alpha_j \right) = I_{\frac{1}{2} + \gamma} \left( \sum_{j \notin w} \alpha_j, \sum_{i \in w} \alpha_j \right)$$

using $\boldsymbol{I}_{a_i - \max_{j \neq i} a_j \leq 2\gamma} \leq \boldsymbol{I}_{a_i - \sum_{j \neq i} a_j \leq 2\gamma} = \boldsymbol{I}_{\sum_{j \neq i} a_j \geq \frac{1}{2} - \gamma}$ for $\boldsymbol{a} \in \Delta^c$ (with equality for $c = 2$ classes), and that $I_z(a, b)$ is the CDF of a Beta distribution with parameters $(a, b)$. $\qquad\square$

*Proof of Theorem 4.* Define $\gamma_2 > \gamma_1$ such that $\gamma := \gamma_2 - \gamma_1$, and $\boldsymbol{\alpha} = K\boldsymbol{\theta}$. From the trivial inequality $\boldsymbol{I}_{x \in A} - \boldsymbol{I}_{x \in B} \leq \boldsymbol{I}_{x \in A} \boldsymbol{I}_{x \notin B}$ we derive

$$\Delta := \ell_{\gamma_1}(\boldsymbol{\theta}, x, y) - \mathbb{E}_{\boldsymbol{\xi} \sim \mathrm{Dir}(\boldsymbol{\alpha})} \ell_{\gamma_2}(\boldsymbol{\xi}, x, y) = \mathbb{E}_{\boldsymbol{\xi} \sim \mathrm{Dir}(\boldsymbol{\alpha})} [\boldsymbol{I}_{M(\boldsymbol{\theta}, x, y) \leq 0} - \boldsymbol{I}_{M(\boldsymbol{\xi}, x, y) \leq \gamma}]$$

$$\leq \mathbb{E}_{\boldsymbol{\xi} \sim \mathrm{Dir}(\boldsymbol{\alpha})} [\boldsymbol{I}_{M(\boldsymbol{\theta}, x, y) \leq 0} \boldsymbol{I}_{M(\boldsymbol{\xi}, x, y) > \gamma}] \leq \mathbb{E}_{\boldsymbol{\xi} \sim \mathrm{Dir}(\boldsymbol{\alpha})} [\boldsymbol{I}_{M(\boldsymbol{\xi}, x, y) - M(\boldsymbol{\theta}, x, y) > \gamma}]$$

$$= \mathbb{P}_{\boldsymbol{\xi} \sim \mathrm{Dir}(\boldsymbol{\alpha})} \left\{ \sum_{i:h_i(x)=y} \xi_i - \max_{j' \neq y} \sum_{i:h_i(x)=k'} \xi_i - \sum_{i:h_i(x)=y} \theta_i + \max_{j \neq y} \sum_{i:h_i(x)=k} \theta_i > 2\gamma \right\}$$

$$\leq \mathbb{P}_{\boldsymbol{\xi} \sim \mathrm{Dir}(\boldsymbol{\alpha})} \left\{ \sum_{i:h_i(x)=y} \xi_i - \sum_{i:h_i(x)=k} \xi_i - \sum_{i:h_i(x)=y} \theta_i + \sum_{i:h_i(x)=k} \theta_i > 2\gamma \right\}$$

where in the last inequality we set $k = \mathrm{argmax}_{k \neq y} \sum_{i:h_i(x)=k} \theta_i$, and use that $\max_j \sum_{i:h_i(x)=j} \theta_i - \max_j \sum_{i:h_i(x)=j} \xi_i \leq \max_j \sum_{i:h_i(x)=j} \theta_i - \sum_{i:h_i(x)=k} \xi_i$ for any $k$. We rewrite the above in vector form (with inner product denoted $\boldsymbol{u} \cdot \boldsymbol{v}$) as

$$\Delta \leq \mathbb{P}_{\boldsymbol{\xi} \sim \mathrm{Dir}(\boldsymbol{\alpha})} \left\{ \underbrace{\frac{1}{\sqrt{2}} \begin{bmatrix} 1 \\ -1 \\ 0 \end{bmatrix}}_{\boldsymbol{u}} \cdot \left( \underbrace{\begin{bmatrix} \sum_{i:h_i(x)=y} \xi_i \\ \sum_{i:h_i(x)=k} \xi_i \\ \sum_{i:h_i(x) \notin \{k,y\}} \xi_i \end{bmatrix}}_{\tilde{\boldsymbol{\xi}}} - \underbrace{\begin{bmatrix} \sum_{i:h_i(x)=y} \theta_i \\ \sum_{i:h_i(x)=k} \theta_i \\ \sum_{i:h_i(x) \notin \{k,y\}} \theta_i \end{bmatrix}}_{\mathbb{E}\tilde{\boldsymbol{\xi}}} \right) > \sqrt{2}\gamma \right\}$$

$$= \mathbb{P}_{\tilde{\boldsymbol{\xi}} \sim \mathrm{Dir}(\tilde{\boldsymbol{\alpha}})} \left\{ \boldsymbol{u} \cdot (\tilde{\boldsymbol{\xi}} - \mathbb{E}\tilde{\boldsymbol{\xi}}) > \sqrt{2}\gamma \right\}$$

where by the aggregation property of the Dirichlet distribution $\tilde{\boldsymbol{\xi}} \sim \mathrm{Dir}(\tilde{\boldsymbol{\alpha}})$ with

$$\tilde{\boldsymbol{\alpha}} := \left[ \sum_{i:h_i(x)=y} \alpha_i, \sum_{i:h_i(x)=k} \alpha_i, \sum_{i:h_i(x) \notin \{k,y\}} \alpha_i \right]^T.$$

Applying Theorem 5 we obtain $\Delta \leq e^{-4(\sum_i \tilde{\alpha}_i + 1)\gamma^2} = e^{-4(\sum_{i=1}^d \alpha_i + 1)\gamma^2}$. This gives the first inequality by setting $\gamma_1 = 0, \gamma_2 = \gamma$. Setting $\gamma_1 = \gamma, \gamma_2 = 2\gamma$ and swapping $\boldsymbol{\theta}$ and $\boldsymbol{\xi}$ gives an almost identical proof (with some signs reversed) of the second inequality. $\qquad\square$

# 4 Empirical evaluation

In this section we empirically validate our results against existing PAC-Bayesian and margin bounds on several classification datasets from UCI (Dua and Graff, 2017), LIBSVM[1] and Zalando (Xiao et al., 2017). Since our main result in Theorem 2 is not associated with any particular algorithm, we use $\boldsymbol{\theta}$ outputted from PAC-Bayes-derived algorithms to evaluate this result

---
[1] https://www.csie.ntu.edu.tw/~cjlin/libsvm/

against other margin bounds (Figure 1) and PAC-Bayes bounds (Figure 2). We then compare optimisation of our secondary result Theorem 3 with optimising those PAC-Bayes bounds directly (Figure 3). All generalisation bounds given are evaluated with a probability $1-\delta=0.95$. Further details not provided here including tabulated results, description of datasets, training mechanisms and compute are provided in Appendix C. The code for reproducing the results is available at https://github.com/vzantedeschi/dirichlet-margin-bound.

**Strong and weak voters.** Similarly to Zantedeschi et al. (2021) we consider both using data-independent and data-dependent voters. This brings our experimental setup in line with a common workflow for machine learning practicioners: the training set is sub-divided into a set for training several different strong algorithms, and a second set on which the weightings of these are optimised. More specifically, the weak voter setting, used only for binary classification, uses axis-aligned decision stumps (denoted *stumps*), with thresholds evenly spread over the input space (6 per feature and per class). The stronger voters (denoted *rf*) are learned from half of the training data, while the other half is used for evaluating and optimising the different generalisation bounds (note this reduces $m$). These take the form of random forests (Breiman, 2001) of $M=10$ trees optimising Gini impurity score on $\frac{n}{2}$ bagged samples and $\sqrt{d}$ drawn features for each tree, with unbounded maximal depth.

**Optimising $\gamma$ and $K$ in bounds.** In reporting margin bounds we optimise over a grid of margin $\gamma$ values in $(0, \frac{1}{2})$, and additionally over $K$ for Theorem 2. Since Theorem 2 and Equation (3) as stated require a fixed margin, we apply a union bound over the values in the grid, replacing $\delta$ in these bounds with $\delta/N$ where $N$ is the number of grid points.

**Existing PAC-Bayes bounds.** We compare to state-of-the-art PAC-Bayesian bounds (and derived algorithms) for weighted majority vote classifiers: the First Order (Langford and Shawe-Taylor, 2003), the Second Order (Masegosa et al., 2020), Binomial (Lacasse et al., 2010) (with the number of voters set to 100) and the two Chebyshev-Cantelli-based (Wu et al., 2021) empirical bounds from categorical-type Gibbs classifiers with parameter $\boldsymbol{\theta}$, and we refer to these as *FO*, *SO*, *Bin*, *CCPBB* and *CCTND* respectively (more details are given in Appendix C). We denote by *f2* the factor two bound derived in Zantedeschi et al. (2021, Annex A.4) from Dirichlet majority votes. All prior distributions for PAC-Bayes bounds, including ours, are set to uniform. We also refer by the same names to the outputs of optimising these bounds with stochastic gradient descent; details on training and initialisation are given in Appendix C.

**Description of figures.** In Figure 1 we compare Theorem 2 with the existing margin bound of Equation (2) and the improved Biggs and Guedj (2022b) bound given in Equation (3). Since Equation (3) is strictly better than the original result and the latter was vacuous in almost all cases considered (see Appendix B), we do not include it. All datasets are for binary classification as the existing results only cover this case, and the $\boldsymbol{\theta}$ values considered are the outputs of either the FO- or f2-optimisation using either the weak or the strong voters described above. Figure 2 extends this evaluation of Theorem 2 to improve generalisation results, by applying it to the models optimised with the PAC-Bayes bounds *FO*, *SO*, *Bin* and *f2* as objective. In this case, we consider both binary and multiclass datasets. In Figure 3 we directly compare the outputs of optimising state-of-the-art PAC-Bayesian bounds with our optimisation-ready variant result Theorem 3. These experiments were carried out on strong voters, as standard in the literature (*e.g.* Lorenzen et al., 2019; Masegosa et al., 2020; Wu et al., 2021).

## 5 Discussion and conclusion

We observe overall that in many cases the existing margin and PAC-Bayes bounds are insufficient to explain the generalisation observed, while our new bound is consistently tight, and sometimes sharp (*i.e.* it approaches the true test error).

Figure 1 demonstrates that existing margin bounds can be insufficient to explain the generalisation observed, which could be construed as a null result for the "margins theory". However, our new bound obtains empirically very sharp results in almost all cases, reaffirming to the theory. Note that due to the non-convexity of our bound, the reported values are local minima and can potentially be improved by applying a thorougher search for the optimal $K$, still giving a similarly valid bound. For instance, simply by enlarging the search space for $K$ our bound drops to $0.36 \pm 0.10$ on *ADULT* with decision stumps as voters, beating existing bounds also in this setting. Unlike the existing results, $\boldsymbol{\theta}$ also arises in the complexity (KL divergence) term and so the bound is not equally tight for every $\boldsymbol{\theta}$ at

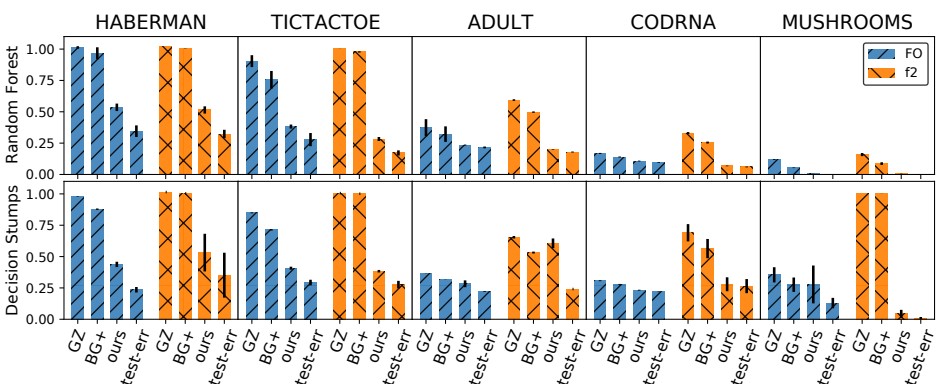

Figure 1: Theorem 2 (**ours**) compared with the margin bounds of Equation (3) (**BG+**), Equation (2) (**GZ**), and the test error. Settings are *rf* (first row) and *stumps* (second row) on the given datasets, with $\theta$ output by optimising either *FO* or *f2* (first and second column groupings respectively).

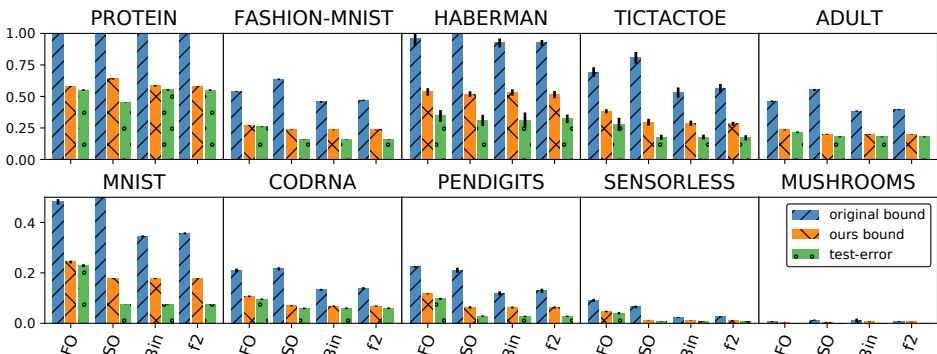

Figure 2: Theorem 2 (*our bound*) compared with the bounds of *FO*, *SO*, *Bin* or *f2* (*original bound*), and test errors. For each column grouping, $\theta$ is the output from optimising the corresponding PAC-Bayes bound (as named underneath) for *rf* on the given dataset. The blue column is the final value of the bound used as objective, the green is the test error, and the orange is the value of our bound when $\theta$ is plugged into it (so that our bound is not used as an objective here).

fixed margin loss. Further examination of this property could add additional nuance and perspective to the theory.

When comparing to existing PAC-Bayes bounds in Figure 2, remarkably Theorem 2 is *always* tighter than just using the bound which is being optimised. We speculate that this arises partially due to the irreducible factors appearing in those bounds; for example the *FO* or *f2* bounds can never be tighter than twice the train loss of the associated Gibbs classifier, while ours has no such limitation. This result is quite valuable as it demonstrates that Theorem 2 can be readily used in an algorithm-free manner: the choice of learning algorithm is up to the practitioner, but the bound will then often provide an excellent guarantee on the obtained weights $\theta$.

Finally, in Figure 3, our optimisation-friendly variant bound Theorem 3 is seen to be competitive in terms of test error while giving an improved-or-equal final bound on all datasets. When considering the less-common setting of binary stumps (see Appendix C) we found that sometimes this objective converged to a sub-optimal local minimum. We speculate that this arises due to the highly non-convex nature of the objective combined with a strong $K$-inflating gradient signal from the $O(e^{-K\gamma^2})$ term. Thus future work to improve these results even further could start with the use of the quasi-convex small-kl relaxation from Thiemann et al. (2017). We note however that this is overall less important than our main results, as both our bounds are still extremely tight when used in an algorithm-free way and applied to the output of another algorithm as discussed above.

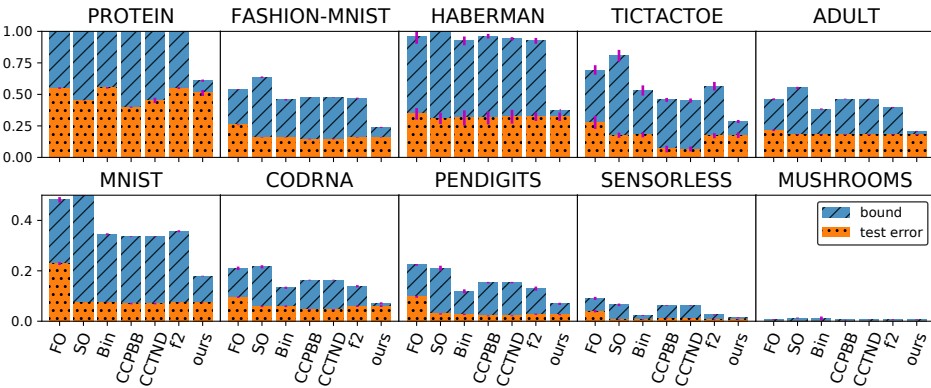

Figure 3: Theorem 3 (**ours**) as optimisation objective compared to other PAC-Bayes results (*FO*, *SO*, *Bin*, *CCPBB* and *CCTND*) as objectives in the *rf* setting. For each objective the test error and bound associated with the objective is shown.

Overall, we note that in many cases (a majority in Figure 2) our main bound of Theorem 2 is very close to the test set bound and thus cannot actually be improved any further, with the problem of providing sharp guarantees based on the training data alone effectively solved in many cases.

**Conclusion.** We obtain empirically very strong generalisation bounds for voting classifiers using margins. We believe these are highly relevant to the community, since voting-based classifiers and margin-maximising algorithms are among the most popular and influential in machine learning. Dirichlet majority votes have already obtained excellent results in the stochastic setting (Zantedeschi et al., 2021), but our new result in Theorem 4 showing they are well-approximated by their mean should open new directions in the more conventional deterministic setting.

Our results also have practical relevance: for example, in the strong voter machine learning workflow described above, instead of setting data aside as a test set, this data can be freed up to learn even stronger voters, since a strong out-of-sample ensemble guarantee can still be provided even *without* a test set.

In future work we hope to expand these results further to other (non-majority) voting schemes like those with score-output voters (as in *e.g.* Schapire et al., 1998), and ensembles of voters with finite VC dimension.

### Acknowledgements

The experiments presented in this paper were carried out using the Grid'5000 testbed, supported by a scientific interest group hosted by Inria and including CNRS, RENATER and several Universities as well as other organizations (see https://www.grid5000.fr). F.B. acknowledges the support of the EPSRC grant EP/S021566/1. V.Z. contributed to this work while being supported from the French National Agency for Research, grant ANR-18-CE23-0015-02. B.G. acknowledges partial support by the U.S. Army Research Laboratory, U.S. Army Research Office, U.K. Ministry of Defence and the U.K. Engineering and Physical Sciences Research Council (EPSRC) under grant number EP/R013616/1; B.G. also acknowledges partial support from the French National Agency for Research, grants ANR-18-CE40-0016-01 and ANR-18-CE23-0015-02.

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
