# A  Properties of the Dirichlet distribution

The Dirichlet measure has probability density function w.r.t. Lebesgue measure given by:

$$f(x_1, \ldots, x_d; \boldsymbol{\alpha}) = \frac{1}{B(\boldsymbol{\alpha})} \prod_{i=1}^{d} x_i^{\alpha_i - 1}$$

where $B(\boldsymbol{\alpha})$ is the multivariate Beta function,

$$B(\boldsymbol{\alpha}) := \frac{\prod_{i=1}^{d} \Gamma(\alpha_i)}{\Gamma(\sum_{i=1}^{d} \alpha_d)}.$$

The mean of a Dirichlet is $\mathbb{E}_{\xi \sim \text{Dir}(\boldsymbol{\alpha})} \boldsymbol{\xi} = \boldsymbol{\alpha} / \sum_{i=1}^{d} \alpha_i$.

The KL divergence between two Dirichlet distributions is the following, given in *e.g.* Zantedeschi et al. (2021):

$$\mathbb{D}_{\text{Dir}}(\boldsymbol{\alpha}, \boldsymbol{\beta}) = \log \frac{B(\boldsymbol{\beta})}{B(\boldsymbol{\alpha})} + \sum_{i=1}^{d} (\alpha_i - \beta_i)(\psi(\alpha_i) - \psi(\alpha_0)) = \log B(\boldsymbol{\beta}) - \mathbb{H}_{\text{Dir}}(\boldsymbol{\alpha}).$$

# B  Additional details on margin bounds

Here we first note the original result from Biggs and Guedj (2022b) that is adapted in Equation (3); since this is obtained by applying an upper bound to the inverse small-kl and an additional step, it is strictly looser than the result we give in Equation (3). Biggs and Guedj (2022b) also uses a dimension doubling trick to allow negative weights (as they consider only the binary case), which we remove here to replace the factor $\log(2d)$ by $\log d$.

**Theorem 6.** *For any margin $\gamma > 0$, $\delta \in (0, 1)$, sample size $m \in \mathbb{N}$, each of the following results holds with probability at least $1 - \delta$ over the sample $S \sim \mathcal{D}^m$ simultaneously for any $\boldsymbol{\theta} \in \Delta^d$,*

$$L_0(\boldsymbol{\theta}) \leq \hat{L}_\gamma(\boldsymbol{\theta}) + \sqrt{\frac{C}{m} \cdot \hat{L}_\gamma(\boldsymbol{\theta})} + \frac{C + \sqrt{C} + 2}{m}, \tag{5}$$

*where $C := 2 \log(2/\delta) + \frac{19}{4} \gamma^{-2} \log d \log m$.*

## B.1  Definition of the margin

We here note that the definition of the margin given in Gao and Zhou (2013) and Biggs and Guedj (2022b) is slightly different from our own, leading to a scaling of the margin definition by a factor of one-half. We show this below.

Both the above papers consider prediction functions like $F_{\boldsymbol{\theta}}(x) = \sum_{i=1}^{d} \theta_i h_i(x)$ with output set $\mathcal{Y} = \{+1, -1\}$. The functions $h_i(x)$ can be positive or negative. The margin is defined as as $y F_{theta}(x)$. We translate this into our equivalent but scaled version as follows:

$$y F_{\boldsymbol{\theta}}(x) = y \left( \sum_{i:h_i(x)=1} \theta_i - \sum_{i:h_i(x)=-1} \theta_i \right) = \sum_{i:h_i(x)=y} \theta_i - \sum_{i:h_i(x)=-y} \theta_i \tag{6}$$

which is double the margin as we define it. Thus $\ell_\gamma(\boldsymbol{\theta}, x, y) = \boldsymbol{I}_{M(\boldsymbol{\theta}, x, y) \leq \gamma} = \boldsymbol{I}_{yF_{\boldsymbol{\theta}} \geq 2\gamma}$ and the condition on the margin $y F(x) \geq \sqrt{8/d}$ given in Gao and Zhou (2013) translates to $M(\boldsymbol{\theta}, x, y) > \sqrt{2/d}$ as we give.

## B.2  Proof of Theorem 6 and Equation (3)

For completeness we provide here short proofs of Equation (3) and Theorem 6. The central proposition used in Biggs and Guedj (2022b) to prove their margin bound and these results for voting algorithms is the following, proved implicitly there and here adapted to our setting.

**Theorem 7** ([Biggs and Guedj](2022b)). *Let $\boldsymbol{\theta} \in \Delta^d$ and define $\rho = \text{Categ}(\boldsymbol{\theta})$ and $\boldsymbol{i} \sim \rho^T$ as $T$ i.i.d. samples from $\rho$ indexed by $j \in [T]$. Then for any $\gamma > 0, T \in \mathbb{N}_+$ and $(x, y \in \{+1, -1\})$,*

$$\ell_0(\boldsymbol{\theta}, x, y) \leq \mathbb{E}_{\boldsymbol{i} \sim \rho^T} \ell_\gamma^C(\boldsymbol{i}, x, y) + e^{\frac{1}{2}T\gamma^2}$$

$$\mathbb{E}_{\boldsymbol{i} \sim \rho^T} \ell_\gamma^C(\boldsymbol{i}, x, y) \leq \ell_\gamma(\boldsymbol{\theta}, x, y) + e^{\frac{1}{2}T\gamma^2}$$

*where we have defined the margin loss for a sum of Categoricals as $\ell_\gamma^{\mathrm{C}}(\boldsymbol{i}, x, y) = \boldsymbol{I}_{yT^{-1} \sum_{t=1}^T h_{i_t}(x) \leq \gamma}$.*

*Proof of Equation* (3). We apply the PAC-Bayes bound Theorem 1 to $\ell_\gamma^C$ with $\rho^T$ as defined in Theorem 7 and a uniform prior of the same form, $\pi^T$. We then substitute the results from Theorem 7 to show that

$$L_0(\boldsymbol{\theta}) \leq \mathrm{kl}^{-1} \left( \hat{L}_\gamma(\boldsymbol{\theta}) + e^{-\frac{1}{2}T\gamma^2}, \ \frac{\mathrm{KL}(\rho^T, \pi^T) + \log \frac{2\sqrt{m}}{\delta}}{m} \right) + e^{-\frac{1}{2}T\gamma^2}.$$

With a uniform prior, $\mathrm{KL}(\rho^T, \pi^T) = T\mathbb{D}_{\mathrm{Cat}}(\theta, d^{-1}\mathbf{1}) \leq T \log d$. Substitution of this upper bound and $T = \lceil 2\gamma^{-2} \log m \rceil$ gives the result. $\qquad\square$

*Proof of Theorem* 6. Beginning with Equation (3), we relax the ceiling using $\gamma \leq \frac{1}{2}$ and $m \geq 2$ for a non-vacuous bound to obtain

$$L_0(\boldsymbol{\theta}) \leq \mathrm{kl}^{-1} \left( \hat{L}_\gamma(\boldsymbol{\theta}) + \frac{1}{m}, \ \frac{C}{2m} \right) + \frac{1}{m}$$

with $C := 2\log(2/\delta) + \frac{19}{4}\gamma^{-2} \log d \log m$. Then using the small-kl upper bound $\mathrm{kl}^{-1}(u, c) \leq u + \sqrt{2cu} + 2c$ we have

$$L_0(\boldsymbol{\theta}) \leq \hat{L}_\gamma(\boldsymbol{\theta}) + \frac{2}{m} + \sqrt{\left( \hat{L}_\gamma(\boldsymbol{\theta}) + \frac{1}{m} \right) \frac{C}{m}} + \frac{C}{m}$$

$$\leq \hat{L}_\gamma(\boldsymbol{\theta}) + \frac{2}{m} + \sqrt{\frac{C}{m} \cdot \hat{L}_\gamma(\boldsymbol{\theta})} + \frac{C + \sqrt{C}}{m}$$

which is the result given. $\qquad\square$

*Proof of Theorem* 7. Using the same method as the beginning of the proof of Equation (6),

$$\ell_0(\boldsymbol{\theta}, x, y) - \mathbb{E}_{\boldsymbol{i} \sim \rho^T} \ell_\gamma^C(\boldsymbol{i}, x, y)$$

$$= \mathbb{E}_{\boldsymbol{i} \sim \rho^T} [\boldsymbol{I}_{yF(x) \leq 0} - \boldsymbol{I}_{yT^{-1} \sum_{t=1}^T h_{i_t}(x) \leq \gamma}]$$

$$\leq \mathbb{P}_{\boldsymbol{i} \sim \rho^T} \left( \frac{1}{2} y(F(x) - T^{-1} \sum_{t=1}^T h_{i_t}(x)) > \frac{1}{2}\gamma \right)$$

$$\leq \exp \left( -\frac{1}{2}T\gamma^2 \right).$$

In the last line we used Hoeffding's inequality for a sum of $T$ random variables bounded in $[-1, 1]$. The other side follows using an identical method with the margin losses reversed. $\qquad\square$

## B.3 Further improvement to the bound

A question which naturally arises from looking at the proof of Equation (3) and Theorem 6 is whether we can do better by choosing $T$ in a more optimal way, rather than just setting it to $\lceil 2\gamma^{-2} \log m \rceil$. We thus prove a bound here which is valid for the optimal choice of $T$; in practice this is seen to be slightly tighter than Equation (3), although the improvement from Theorem 6 to that result is far greater.

For any $\boldsymbol{\theta} \in \Delta^d$ with probability at least $1 - \delta$ over the sample,

$$.L_0(\boldsymbol{\theta}) \leq \inf_{T \in \mathbb{N}_+} \left[ \mathrm{kl}^{-1}\left( \hat{L}_\gamma(\boldsymbol{\theta}) + e^{-\frac{1}{2}T\gamma^2},\ \frac{T(\log d - \mathbb{H}_{\mathrm{Categ}}[\boldsymbol{\theta}]) + \log \frac{m}{\delta}}{m} \right) + e^{-\frac{1}{2}T\gamma^2} \right] \quad (7)$$

A slightly weaker version of this result, with an extra $m^{-1}\log(2\sqrt{m})$ term, can be proved from

$$L_0(\boldsymbol{\theta}) \leq \mathrm{kl}^{-1}\left( \hat{L}_\gamma(\boldsymbol{\theta}) + e^{-\frac{1}{2}T\gamma^2},\ \frac{T\log d + \log \frac{2\sqrt{m}}{\delta}}{m} \right) + e^{-\frac{1}{2}T\gamma^2},$$

which is shown in the proof of Equation (3). We note that the optimal $T$ depends on the data only through $\hat{L}_\gamma(\boldsymbol{\theta}) \in \{0, m^{-1}, 2m^{-1}, \ldots, 1\}$. The last possibility gives a trivial bound. A union bound over the $m$ non-vacuous possibilities gives Equation (7) with the extra logarithmic factor.

In order to remove this term, we use a slightly more sophisticated argument applied to a different PAC-Bayes bound (Theorem 8) given below. This result uses the function (defined for $C > 0, p \in [0, 1]$)

$$\Phi_C(p) = -\frac{1}{C} \log(1 - p + pe^{-C})$$

which relates to the small KL (Theorem 9).

**Theorem 8** (Catoni (2007))**.** *Given data distribution $\mathcal{D}$ on $\mathcal{X} \times \mathcal{Y}$, prior $P \in \mathcal{M}^1(\mathcal{H})$, $C > 0$ and $\delta \in (0, 1)$, the following hold each with probability $\geq 1 - \delta$ over $S \sim D^m$, for all $Q \in \mathcal{M}^1(\mathcal{H})$*

$$\mathbb{E}_{h \sim Q} L(h) \leq \Phi_C^{-1}\left( \mathbb{E}_{h \sim Q} \hat{L}(h) + \frac{\mathrm{KL}(Q, P) + \log \frac{1}{\delta}}{Cm} \right)$$

**Theorem 9** (Germain et al. (2009), Proposition 2.1)**.** *For any $0 \leq q \leq p < 1$,*

$$\sup_{C > 0} \left[ C\Phi_C(p) - Cq \right] = \mathrm{kl}(q, p).$$

*Proof of Equation* (7)*.* We substitute Theorem 7 into Theorem 8 with the categorical loss and a uniform prior, $\pi^T$. and KL upper bound as in the above proof. as we obtain for any data-independent $C > 0, T \in \mathbb{N}_+, \gamma > 0$ that

$$L_0(\boldsymbol{\theta}) - e^{-\frac{1}{2}T\gamma^2} \leq \Phi_C^{-1}\left( \frac{k}{m} + e^{-\frac{1}{2}T\gamma^2} + \frac{T\log d + \log \frac{1}{\delta}}{Cm} \right).$$

where $k := m\hat{L}_\gamma(\boldsymbol{\theta})$ is the number of margin errors.

Since the only quantity on the left hand side in this bound unknown before we see data is the value of $k$, there exists a $C_k$ dependent on the value of $k$ that optimises the bound, and a $T_k$ that depends on this pair. Since there are only $m$ such values giving non-vacuous bounds ($k = m$ is trivially vacuous), we can apply a union bound over all these bounds with $\delta = \delta/m$ to give the following with probability $\geq 1 - \delta$:

$$L_0(\boldsymbol{\theta}) \leq \min_{T \in \mathbb{N}_+} \min_{C > 0} \left[ e^{-\frac{1}{2}T\gamma^2} + \Phi_C^{-1}\left( \frac{k}{m} + e^{-\frac{1}{2}T\gamma^2} + \frac{T\log d + \log \frac{m}{\delta}}{Cm} \right) \right].$$

Applying the inversion of Theorem 9 gives the second result. $\qquad\square$

## B.4 Comparison of margin bounds

In Figure 4 we compare the various bounds given above in a non-experimental way, fixing the margin loss $\hat{L}_\gamma$ to a particular value and seeing how the bounds change if that value of the loss is achieved for different values of the margin $\gamma \in (0, 0.5)$. Since (uniquely among the bounds), the value of $\boldsymbol{\theta}$ appears in our bound Theorem 2, we show three different sampled possible values, drawn uniformly from the simplex.

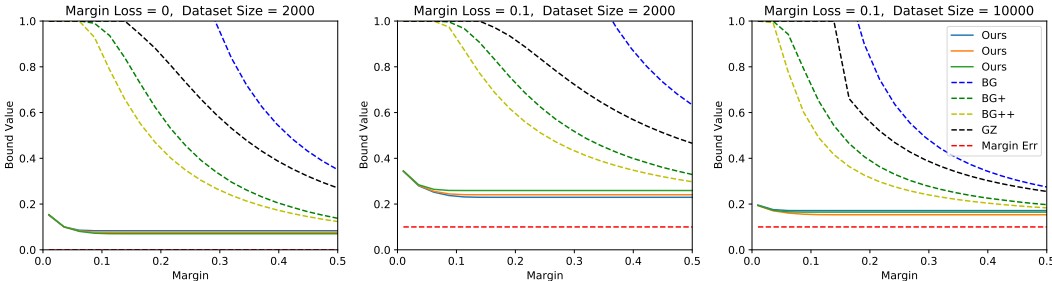

Figure 4: Values of different bounds versus margin at margin error $\hat{L}_\gamma$ (0 or 0.1). Dimension $d = 100$, probability $\delta = 0.5$ and dataset size $m$ (2000 or 10000) are also fixed. The bounds are Theorem 2 (**ours**) with three different samples $\boldsymbol{\theta} \sim \text{Uniform}(\Delta^d)$, compared with the margin bounds of Theorem 6 (**BG**), Equation (3) (**BG+**), Equation (7) (**BG++**), Equation (2) (**GZ**), and the margin error $\hat{L}_\gamma$.

The results for "categorical"-based bounds demonstrate that the refined bounds Equations (3) and (7) are much tighter that the result as given in Theorem 6 by Biggs and Guedj (2022b). Both these refinements are also tighter than Equation (2) from Gao and Zhou (2013). We used Equation (3) in the main paper because it is closer to an exiting result (as it appears in the proof from Biggs and Guedj, 2022b), and is not much worse than the refinement Equation (7), particularly when compared to our far stronger new result Theorem 2.

This figure also shows that, at least for some values of $\boldsymbol{\theta}$, this new bound can be far tighter than all the existing bounds. One interesting facet of this is that the bound is improved very little for $\gamma$ above a certain point, quite a different behaviour to the other bounds. Empirically this was seen too in our other experiments, with the optimised $\gamma$ often being quite small. Of course, for some values of $\boldsymbol{\theta}$ this bound will be weaker, but we observe the same kind of results in our main experimental results, where this is a learned value.

## C  Additional experimental details and evaluations

**Dataset descriptions.**

We provide the description of the classification datasets considered in our empirical evaluation.

> ***Haberman* (UCI)** prediction of survival of $n = 306$ patients who had undergone surgery from $d = 3$ anonymized features.

> ***TicTacToe* (UCI)** determination of a win for player $x$ at TicTacToe game of any of the $n = 958$ board configurations ($d = 9$ categorical states).

> ***Mushrooms* (UCI)** prediction of edibility of $n = 8,124$ mushroom sample, given their $d = 22$ categorical features describing their aspect.

> ***Adult* (LIBSVM a1a)** determining whether a person earns more than 50K a year ($n = 32,561$ people and $d = 123$ binary features).

> ***CodRNA* (LIBSVM)** detection of non-coding RNAs among $n = 59,535$ instances and from $d =$ features.

> ***Pendigits* (UCI)** recognition of hand-written digits (10 classes, $d = 9$ features and $n = 12,992$).

> ***Protein* (LIBSVM)** $d = 357$ features, $n = 24,387$ instances and 3 classes.

> ***Sensorless* (LIBSVM)** prediction of motor condition ($n = 58,509$ instances and 11 classes), with intact and defective components, from $d = 48$ features extracted from electric current drive signals.

> ***MNIST* (LIBSVM)** prediction of hand-written digits ($n = 70,000$ instances and 10 classes) from $d = 28 \times 28$ gray-scale images.

> ***Fashion-MNIST* (Zalando)** prediction of cloth articles ($n = 70,000$ instances and 10 classes) from $d = 28 \times 28$ gray-scale images.

In all experiments, we convert all categorical features to numerical using an ordinal encoder and we standardize all features using the statistics of the training set.

**Baseline descriptions.** We report the generalization bounds of the literature used for training weighted majority vote classifiers in our comparison. We additionally note: $\text{Cat}(\boldsymbol{\theta})$ the categorical distribution over the base classifiers (with $\theta_i$ the weight associated to voter $h_i \in \mathcal{H}$), and $\mathbb{D}_{\text{Cat}}(\boldsymbol{\theta}), \boldsymbol{\pi})$ the KL-divergence between two categorical distribution with parameters $\boldsymbol{\theta}$ and $\boldsymbol{\pi}$; $\ell_{\text{TND}}(h, h', x, y) := \boldsymbol{I}_{h(x) \neq y \wedge h'(x) \neq y}$ the tandem loss proposed in Masegosa et al. (2020) and $\hat{L}_{\text{TND}}(h, h') := \mathbb{E}_{(x,y) \sim \text{Uniform}(S)} \ell_{\text{TND}}(h, h', x, y)$ its in-sample estimate; $\ell_{\text{Bin}}(\boldsymbol{\theta}, N, x, y) := \sum_{k=\frac{N}{2}}^{N} \binom{N}{k} M(\boldsymbol{\theta}, x, y)^k (1 - M(\boldsymbol{\theta}, x, y))^{(N-k)}$ the probability that among $N$ voters randomly drawn from $\text{Cat}(\boldsymbol{\theta})$ at least $\frac{N}{2}$ of them are incorrect, as defined in Lacasse et al. (2010).

- First Order (FO, Langford and Shawe-Taylor, 2003):
  For any $\mathcal{D} \in \mathcal{M}_1^+(\mathcal{X} \times \mathcal{Y})$, $m \in \mathbb{N}_+$, $\delta \in (0, 1)$, and prior $\boldsymbol{\pi} \in \Delta^d$, with probability at least $1 - \delta$ over the sample $S \sim \mathcal{D}^m$ simultaneously for every $\boldsymbol{\theta} \in \Delta^d$,

$$L_0(\boldsymbol{\theta}) \leq 2 \,\text{kl}^{-1} \left( \mathbb{E}_{h \sim \text{Cat}(\boldsymbol{\theta})} \hat{L}_0(h), \ \frac{\mathbb{D}_{\text{Cat}}(\boldsymbol{\theta}, \boldsymbol{\pi}) + \log \frac{2\sqrt{m}}{\delta}}{m} \right).$$

- Second Order (SO, Masegosa et al., 2020):
  For any $\mathcal{D} \in \mathcal{M}_1^+(\mathcal{X} \times \mathcal{Y})$, $m \in \mathbb{N}_+$, $\delta \in (0, 1)$, and prior $\boldsymbol{\pi} \in \Delta^d$, with probability at least $1 - \delta$ over the sample $S \sim \mathcal{D}^m$ simultaneously for every $\boldsymbol{\theta} \in \Delta^d$,

$$L_0(\boldsymbol{\theta}) \leq 4 \,\text{kl}^{-1} \left( \mathbb{E}_{h \sim \text{Cat}(\boldsymbol{\theta}), h' \sim \text{Cat}(\boldsymbol{\theta})} \hat{L}_{\text{TND}}(h, h'), \ \frac{2\mathbb{D}_{\text{Cat}}(\boldsymbol{\theta}, \boldsymbol{\pi}) + \log \frac{2\sqrt{m}}{\delta}}{m} \right).$$

- Binomial (Bin, Lacasse et al., 2010):
  For any $\mathcal{D} \in \mathcal{M}_1^+(\mathcal{X} \times \mathcal{Y})$, $m \in \mathbb{N}_+$, $N \in \mathbb{N}_+$, $\delta \in (0, 1)$, and prior $\boldsymbol{\pi} \in \Delta^d$, with probability at least $1 - \delta$ over the sample $S \sim \mathcal{D}^m$ simultaneously for every $\boldsymbol{\theta} \in \Delta^d$,

$$L_0(\boldsymbol{\theta}) \leq 2 \,\text{kl}^{-1} \left( \mathbb{E}_{(x,y) \sim \text{Uniform}(S)} \ell_{\text{Bin}}(\boldsymbol{\theta}, N, x, y), \ \frac{N\mathbb{D}_{\text{Cat}}(\boldsymbol{\theta}, \boldsymbol{\pi}) + \log \frac{2\sqrt{m}}{\delta}}{m} \right).$$

- Chebyshev-Cantelli tandem loss bound (CCTND, Wu et al., 2021, Theorem 12);
- Chebyshev-Cantelli tandem loss bound with an offset (CCPBB, Wu et al., 2021, Theorem 15);
- Dirichlet Factor-Two (f2, Zantedeschi et al., 2021):
  For any $\mathcal{D} \in \mathcal{M}_1^+(\mathcal{X} \times \mathcal{Y})$, $m \in \mathbb{N}_+$, $\delta \in (0, 1)$, and prior $\boldsymbol{\beta} \in \mathbb{R}_+^d$, with probability at least $1 - \delta$ over the sample $S \sim \mathcal{D}^m$ simultaneously for every $\boldsymbol{\theta} \in \Delta^d$ and $K > 0$,

$$L_0(\boldsymbol{\theta}) \leq 2 \,\text{kl}^{-1} \left( \mathbb{E}_{\boldsymbol{\xi} \sim \text{Dir}(K\boldsymbol{\theta})} \hat{L}(\boldsymbol{\xi}), \ \frac{\mathbb{D}_{\text{Dir}}(K\boldsymbol{\theta}, \boldsymbol{\beta}) + \log \frac{2\sqrt{m}}{\delta}}{m} \right).$$

**Optimisation of PAC-Bayesian bounds.** To optimize the baselines *CCPBB* and *CCTND*, we rely on the code released by its authors [2], with the Gradient Descent option and building random forests as described in our main text. When optimising the PAC-Bayesian bounds *FO*, *SO*, *Bin*, *f2* and ours, we initialize $\boldsymbol{\theta}$'s to be uniform, i.e. $\theta_i = 1/d$, and $K = 2$. We then optimise the posterior parameters of the method ($\alpha = K\theta$ for Dirichlet, and $\theta$ for Categorical distributions) with the Adam optimiser (Kingma and Ba, 2015) with running average coefficients $(0.9, 0.999)$, batch size equal to 100 and learning rate set to $0.1$. All methods are run for a maximum of 100 epochs with patience of 25 epochs for early stopping and a learning rate scheduler reducing it by a factor of 10 with 2 epochs patience.

---

[2] https://github.com/StephanLorenzen/MajorityVoteBounds/tree/44cec987865ddce01cd27076019394538cee85ca/NeurIPS2021

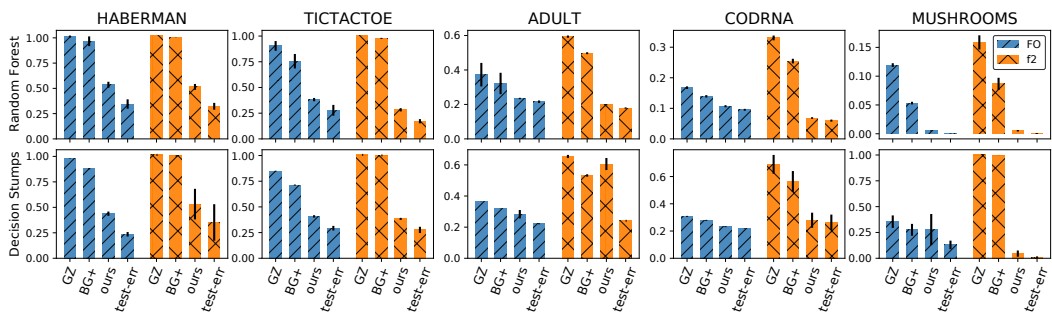

Figure 5: Theorem 2 (**ours**) compared with the margin bounds of Equation (3) (**BG+**), Equation (2) (**GZ**), and the test error. Settings are *rf* (first row) and *stumps* (second row) on the given datasets, with $\theta$ output by optimising either *FO* or *f2* (first and second column groupings respectively).

At each run of an algorithm, we randomly split a dataset into training and test sets of sizes $80\% - 20\%$ respectively, and optimise/evaluate the bounds only with the half of the training set that was not used for learning the voters (in the case of data-dependent ones). Note that we do not make use of a validation set, as we use the risk certificates as estimate of the test error for model selection. Finally, we report the value of Seeger's "small-kl" bound of Theorem 1, even when a different type of bound has been optimised (*e.g.* for the *CCPBB* and *CCTND* baselines), and we average all results over 5 different trials.

**Margin bound comparison.** Given a pre-trained model, hence fixed $\theta$ and initial $K_{init}$ (which is different from 1. only for the models trained via Dirichlet bounds), we search for its optimal risk certificate by evaluating a given bound at $1,000$ values of $\gamma$, spaced evenly on a log scale with base 10 and in the interval $[10^{-4}, 0.5)$. For our margin bound, for each of these $\gamma$ values we also optimise $K \in [K_{init}, K_{init} \, 2^{16}]$ using the golden-section search technique to obtain the tightest upper bound. Notice that this does not add significant computational overhead to the search. Also for these experiments, the bounds are evaluated with the portion of training data that was not used for learning the voters.

**Compute.** All experiments were run on a virtual machine with 16 vCPUs and $128Gb$ of RAM.

## C.1 Additional results

In Figure 5, Figure 6 and Figure 7 we report the results from Figure 1, Figure 2 and Figure 3 in the main text. Here we deploy a different scale per dataset so that they can be easily read, also when the bounds and test errors are very small. Additionally, in Figure 8 we provide the test errors and risk certificates obtained by optimising the generalization bounds with decision stumps as voters. Although our certificates are always the tightest, we found that in some cases our method converges to sub-optimal solutions. We speculate that this arises due to the highly non-convex nature of the objective combined with a strong $K$-inflating gradient signal from the $O(e^{-K\gamma^2})$ term. Thus future work to improve these results even further could start with the use of the quasi-convex small-kl relaxation from Thiemann et al. (2017). We note however that this is overall less important than our main results, as both our bounds are still extremely tight when used in an algorithm-free way and applied to the output of another algorithm.

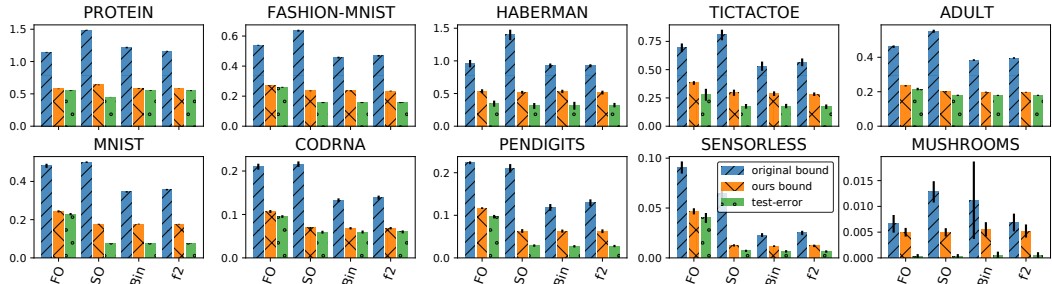

Figure 6: Theorem 2 (*our bound*) compared with the bounds of *FO*, *SO*, *Bin* or *f2* (*original bound*), and test errors. For each column grouping, $\boldsymbol{\theta}$ is the output from optimising the corresponding PAC-Bayes bound for *rf* on the given dataset.

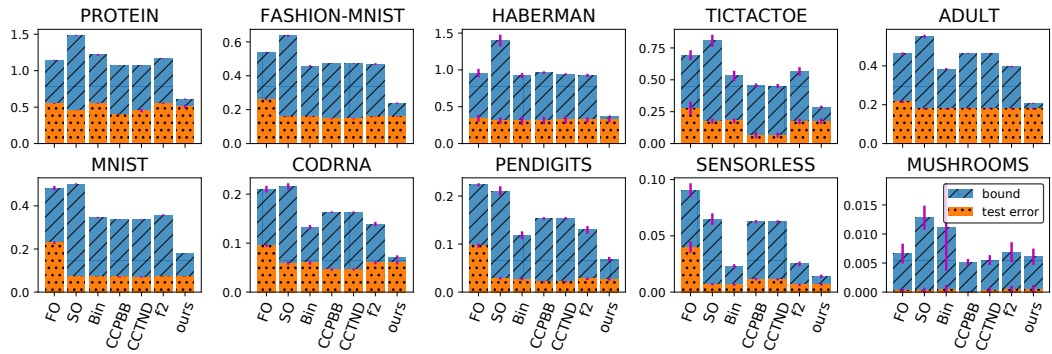

Figure 7: Theorem 3 (**ours**) as optimisation objective compared to other PAC-Bayes results (*FO*, *SO*, *Bin*, *CCPBB*, *CCTND* and *f2*) as objectives, with a Random Forest as set of voters.

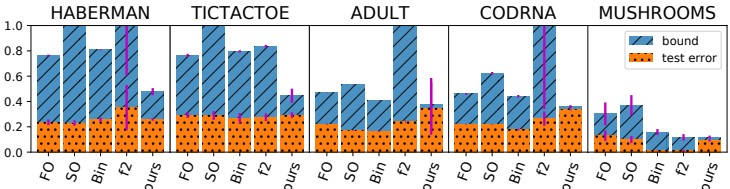

Figure 8: Theorem 3 (**ours**) as optimisation objective compared to other PAC-Bayes results (*FO*, *SO*, *Bin*, *f2*) as objectives, with decision stumps as voters.