# OpenReview forum: "On Margins and Generalisation for Voting Classifiers"
_NeurIPS.cc/2022/Conference — NeurIPS 2022 Accept_

### Official Review · Reviewer_uYQj · 2022-07-11

**Rating:** 7
**Confidence:** 4
**Soundness:** 3 good
**Presentation:** 3 good
**Contribution:** 3 good

**Summary:**

The paper "On Margins and Generalisation for Voting Classifiers" proposes a novel generalization bound for voting classifiers (i.e. ensembles) using a Dirichlet posterior over the ensemble members. The novel bound explores a new direction for generalization bounds for ensembles and also seems to have decent results in practice. The paper is generally well-written and easy to follow. The theoretical contribution is clearly stated and the experimental evaluation seems decent.

**Questions:**

1) In the appendix you write that you initialize $\theta$ uniformly in $[0.01, 1]$ and then optimize over it. I assumed that one would initialize $\theta$ as the weight of each ensemble member for random forests, i.e. $\theta_i = 1/|\mathcal H|$ in the case of random forest. Why do you use different values here? Did I miss something?
2) The results in Fig 1. / 2. are comparably bad to e.g. classic RF. For example, a vanilla RF has a performance around $> 97 $% on mnist whereas you report errors in the range of $10-20 $%. I understand that raw performance is not the main focus of this paper (and hence I dont really care), but I am curios why there is this large performance gap. Do you think this is because $\theta$ is optimized afterwards?
3) I expected a generalization bound to be used as a "plug in" explanation for any voting classifier, e.g. as in Theorem 2. Now I wonder what the impact of $\theta$ is here. From my point of view, $\theta$ are simply the weights of the individual classifiers (e.g. equal for Random Forests or (normalized) floats for boosting). Is there any harm in using Theorem 2 to compute the error of, e.g. AdaBoost and if not, did you try that?

**Limitations:**

There might be some limitations wrt. to $\theta$, but I am not sure about this. Please see my question above.

**Strengths And Weaknesses:**

I like to paper and I do not have much to discuss about it to be honest. The paper combines different aspects from theory into a non-trivial, novel bound. It is well-written and easy to follow. The quality of the math seems sound, although I admit that I did only skimmed the appendix. I believe the use of the Dirichlet posterior is an interesting idea and can be used to as a starting point for future research. I have some minor remarks regarding the references that can easily be fixed for a camera ready and some question wrt. to the weights of each member in the ensemble (see below). In short:

- (+) Novel and non-trivial combination of different results from literature
- (+) Generally well-written and comparably easy to follow
- (+) Decent empirical evaluation
- (-) High test error in the empirical evaluation (see my question below)
- (-) The citations should be improved for more consistency. I suggest to double-check references {1,14,16,17,23,24,26,36,47} via https://dblp.uni-trier.de/ and copy/paste the extended Bibkeys

---

> ### Author Response · Authors · 2022-08-02
> **Response to reviewer uYQj**
>
> We thank the reviewer again for the strong support shown for our paper. Please see also our general response.
>
> We will tidy up the references (thanks for pointing these out) and incorporate answers to the questions below:
>
> 1. We used this initialisation scheme to replicate the setup in Zantedeschi et al. (2021). However, re-running experiments with the well-motivated uniform initialisation you suggested actually leads to improved results! Both test error and bound are improved in a variety of cases so we will change all results to this scheme for the revision. Thank you for this suggestion!
>
> 2. Using the sklearn random forest on MNIST (as in, e.g. [this kaggle example](https://www.kaggle.com/code/ashwani07/mnist-classification-using-random-forest/notebook)) by default uses 100 trees, while we use only 10. We also differ in that the trees are trained using half of the dataset rather than full-dataset bag sizes. In this setup, random forest (with uniform $\theta$) attains an error of 0.073 with 10 trees which is improved by most of the PAC-Bayes algorithms we study, with the exception of FO, which attains a (significantly worse) error of around 20%. This occurs because FO tends to put most of its weight on a single best tree (as observed by e.g. Wu et al., 2021, and Masegosa et al., 2020). We will be happy to report also our bound values for this vanilla random forest in Figure 1.
>
> 3. You are correct that our bound in Theorem 2 can be used as a plug-in for any value of $\theta$. However the classifiers/voters being weighted must be fixed based on a subset of the training set, whereas using Adaboost each new voter is chosen based on the entire dataset. Therefore some additional form of capacity control on these voters would be needed: if they overfit the training set generalisation may not be possible.
> An extension to our work could consider freely choosing the voters from a class of fixed VC dimension (as in Schapire et al.), allowing bounds for e.g. boosting algorithms. Extending PAC-Bayes bounds to general VC classes is highly technical (see e.g. [1]), but is an important direction for future work.
>
> [1] PAC-Bayes, MAC-Bayes and Conditional Mutual Information: Fast rate bounds that handle general VC classes. Grunwald, Steinke, Zakynthinou, 2021.

---

> > ### Comment · Reviewer_uYQj · 2022-08-08
> > **Response to Response to reviewer uYQj**
> >
> > Thank you for answering my question and I am glad that my suggestion could improve the results! I am happy to see this paper accepted.

---

### Official Review · Reviewer_hEGP · 2022-07-14

**Rating:** 7
**Confidence:** 3
**Soundness:** 4 excellent
**Presentation:** 4 excellent
**Contribution:** 3 good

**Summary:**

This paper deals with the theory of (multi-category) pattern classification. The authors introduce a new PAC-Bayes bound for majority voting of finite ensembles of classifiers. It is based on Dirichlet posteriors. A variant is proposed to serve as the objective function of a training algorithm (train the model itself). Those two bounds are compared with the state-of-the-art ones in the framework of an empirical study.

**Questions:**

A few typos should be corrected. Two examples :
-line 93 : complimentary -> complementary
-line 470 : pac-bayesian -> PAC-Bayesian

**Strengths And Weaknesses:**

The contribution appears to bring a significant improvement compared to the state of the art, even though it is not that original, since it is close to a result by Zantedeschi et al. (2021). I found it technically sound. It is noticeable that the paper is very well written (clear).

---

> ### Author Response · Authors · 2022-08-02
> **Response to reviewer hEGP**
>
> We thank the reviewer again for the positive evaluation of our paper. Please see also our general response.
>
> Concerning the originality of the contribution, while the use of Dirichlet posteriors has been introduced in Zantedeschi et al. (2021), their de-randomisation using margins is completely novel. The main technical result (Theorem 4) that is used for this combines the idea of sub-Gaussian concentration (more common in margins literature) with the special aggregation properties of the Dirichlet distribution. This results in a de-randomisation step which surprisingly does not depend on the number of base voters, a limitation of previous works. The tightness of this step leads to the great improvement in risk certificates compared with prior work.

---

### Official Review · Reviewer_KkjL · 2022-07-26

**Rating:** 7
**Confidence:** 3
**Soundness:** 3 good
**Presentation:** 3 good
**Contribution:** 4 excellent

**Summary:**

On Margins and Generalisation for Voting Classifiers

In this paper, the authors take a generalization bound for stochastic majority votes from prior work and show how to derive a new margin bound for majority vote algorithms using Dirichlet priors. As stepping stone they prove a new result relating the misclassification loss to the stochastic voting loss. This bound is empirically verified to have better characteristics in that it comes closer to accurately predicting test set error than related work. The authors also derive an optimizable target version of their bound which appears to yield good results compared to other similar approaches in the literature on a variety of datasets.

**Questions:**

Two minor points: I would have liked to see a precise statement of Zantedeschi et al's bound in section 2.3. In particular, I missed the intuition behind "the use of such Dirichlet distributions...allows the correlation between voters to be more carefully considered."
Second, I missed why in figure 2 the "our bound" results change for every baseline.

**Limitations:**

There is no specific limitations section, but the authors do describe gaps between their results and empirical results as well as cases when their bounds may not be accurate (albeit very briefly)

**Strengths And Weaknesses:**

This paper, although very dense, is making important contributions to the understanding of ensemble classifiers, in particular to the margin argument of why such classifiers generalize. It struck me as very insightful to be able to use the stochastic voting bound to prove the majority voting result. That the result also turns out to tighter than prior work as well as empirically perform well makes it very strong (and cool!).
On the whole, I think this is a strong paper that is using very insightful connections from recent prior work to improve the state of the art generalization bounds on voting classifiers.

---

> ### Author Response · Authors · 2022-08-02
> **Reply to reviewer KkjL**
>
> We thank the reviewer again for the strong support shown for our contributions. Please see also our general response.
>
> We note that the exact expression of the bound in Zantedeschi et al. (2021) is included in the appendix (l. 661) due to space considerations, but we will be glad to include it in the main paper in case of acceptance with the additional space given.
>
> In Figure 2, we evaluate our bound for the different models learned by each baseline. So the $\theta$ used is different in each case and minimises the baseline bound (given in blue) and has given test error (green). Applying our bound Th. 2 in a plug-in fashion to this $\theta$ gives the bounds in orange, which vary as they depend on the different $\theta$ values used, which have different training errors and generalisation capabilities. Remarkably, our bound gives tighter values here than the bounds which have been directly optimised, likely due to their loose de-randomisation.
>
> ----
>
> Apologies, we now realise we missed one of your questions: "the use of such Dirichlet distributions...allows the correlation between voters to be more carefully considered." This comes from Zantedeschi et al., where the idea is that with a categorical distribution the Gibbs risk is simply an average of the losses of individual predictors, without talking into account how well the combination of their predictions performs. Conversely, the Dirichlet distribution gives a (stochastic) majority vote of predictors, so if the errors of base voters are de-correlated, the better performance that results can be accounted for in the bound.

---

### Author Response · Authors · 2022-08-02
**General response**

We warmly thank all the reviewers for their time, positive evaluation and useful feedback that is invaluable to improve the quality of our manuscript.

We are glad all reviewers found our contributions significant and appreciated the clarity and soundness of our analyses. Reviewer KkjL noted the “important contributions to the understanding of ensemble classifiers” and called our majority vote result “very insightful”, noting that overall our results “turn[s] out tighter than prior work as well as empirically perform[s] well makes it very strong (and cool!)”. Reviewer hEGP noted that empirical results are  “significant improvement compared to the state of the art” and called the manuscript “very well written (clear)”. This sentiment was shared by Reviewer uYQj calling it “well-written and easy to follow”, noting that the work “combines different aspects from theory into a non-trivial, novel bound”.

As an additional note to all reviewers, we would like to emphasise that our empirical results are not only a considerable improvement over existing ones but in many cases are effectively sharp, i.e. tightly bound the true test error. Therefore they cannot actually be further improved. We see our work therefore not only as a theoretical contribution, but one which could in certain cases be practically used, since it may be possible to avoid the use of a hold-out test set entirely. This would allow more data to be allocated to training.


-----

UPDATE: We thank all the reviewers again for their helpful comments. We have uploaded a minor revision with the margin bound figures updated to the new initialisation suggested by Reviewer uYQj. Additional commentary taking up additional space will be added in the camera ready in case of acceptance.

---

### Meta-Review · Area_Chair_PkPR · 2022-08-30

**Recommendation:** Accept
**Confidence:** Certain

**Metareview:**

All reviewers uniformly agree on the paper being interesting and worth publishing -- a very fine read. While the authors have already uploaded an updated version of their paper with minor revisions, I encourage them to use the camera-ready version to carry further improvements taking into accounts all reviews.



**Award:**

No

---

### Decision · Program_Chairs · 2022-09-14

Accept